# Analysis of Online Classes in Physical Education during the COVID-19 Pandemic

**Jieun Yu [1] and Yongseok Jee [2],***

1    Department of Physical Education, Korea University, Seoul 02841, Korea; jieun12155@naver.com
2    Department of Leisure and Marine Sports, Hanseo University, Seosan 31962, Korea
*   Correspondence: jeeys@hanseo.ac.kr; Tel.: +82-41-660-1028

**Abstract:** Background and objectives: This study analyzed the effectiveness of the online practical classes (OPC) in physical education (PE) in compliance with the ADDIE model during the COVID-19 pandemic. Materials and Methods: Participants had no prior experience in OPC and total 75 participants were enrolled in this study. This study selected 15 universities in consideration of regional equality and randomly selected two professors and three students from each university. Results: (1) The learning interventions were not feasible for team projects. (2) In the implementation phase, most learners felt that errors persisted. (3) In the evaluation phase, educators reported unenthusiastic involvement of students and the learners were merely submitting assignments. (4) An appropriate level of the effectiveness through OPC showed significantly different between educators and learners. Conclusions: The findings indicate that timely and quality feedback should be provided for the successful execution of OPC in PE; the educators should prepare ahead and reduce technical errors and motivate learners continuously. Lastly, to prepare for the new normal after COVID-19, universities should provide enough time for educators to make OPC-videos and teach students in real time to ensure consistent feedback.

**Keywords:** online practical classes; ADDIE model; synchronous online lecture; feedback

## 1. Introduction

The 2019 COVID-19 is an infectious disease caused by severe acute respiratory syndrome coronavirus. The COVID-19 pandemic has restricted the human-to-human contact [1], and social distancing is regarded as the most effective preventive strategy because there is neither a specific treatment, nor has a vaccine for COVID-19 been discovered as yet [2], thus affecting the educational environment. On 23 February 2020, the South Korean government accentuated the response to the COVID-19 pandemic to a serious level. On March 2, the Korean Ministry of Education announced the 'Operation and Support Plan for Education', which contains the operation of the university's first semester scheduled from the beginning of March to the middle of June (15 weeks) in 2020. In response to this plan, most universities started operating online classes and seminars (Webinars) instead of conducting face-to-face lectures.

In fact, the demand for the conversion of existing face-to-face classes into online classes is not a temporary phenomenon in the educational field. Due to the increased interest in and acceptance of online classes, professors are being asked to design online classes [3]. Online learning is reported to be a popular alternative teaching method in many institutional organizations all over the world [4]. Specifically, the enrolment in online classes was reported to increase from 2 million in 2003 to 6.7 million in 2014 [5]. However, online practical classes (OPC) in physical education (PE) are not easy to teach or learn for educators and students, respectively. Many studies have reported that the interaction between the students and the educator in online courses is inferior to that in the traditional classroom setting, making student engagement difficult [6,7]. Especially, practical exercises in PE comprise subjects that move the body, improve the individual

health and/or physical skills, reduce stress, and overcome mental limitations [8,9]. In fact, PE classes help individuals to learn motor skills [10], promote collaboration among students, or encourage social interactions through group events or competitive events [11]. However, online PE classes may not provide these educational benefits to the students because the educator and the students are physically and spatially separated. Nevertheless, during the COVID-19 pandemic situation, it is imperative to move online in undergraduate PE.

With the growth of online learning or instruction, a pedagogically effective instructional design model for online education is strongly required to facilitate the development and delivery of engaging online learning environments because a poorly designed online lecture makes learners get lost, loses their interest and feel distressed [12]. Instructional design or instructional systematic design is defined as "the practice of creating instructional experiences which make the acquisition of knowledge and skill more efficient, effective, and appealing", and an effective instructional design is very beneficial for instructors and learners [12,13]. The ADDIE model is one of frequently cited instructional design models, which is utilized to create training programs in a steady and solid form in diverse fields [14–16]. The acronym ADDIE denotes the following five phases: Analysis, design, development, implementation, and evaluation [17]. The analysis phase in which educators ascertain the needs of the learners involves setting educational goals and determining what needs to be taught. During the analysis phase, educators can use various methods including surveys, expert consensus, and opinions of graduates of the program to gather the information. In design phase, educators describe how to deliver their instruction to meet the objectives under a broad overview or blueprint. The design phase includes selecting the optimal methods and creating useful learning goals. The component of the instruction is concretely planned in the development phase, and then educators deliver the instruction in the implementation. Finally, in the evaluation phase, educators get feedback on the instruction or lecture and appropriately adjust it to their program of instruction. One of important methods evaluating the educational effectiveness is to objectively measure how well the learners have acquired the knowledge, skills or attitudes being taught [17].

The ADDIE model is fundamental in that all the components of the instructional process react to any stimulus, thus helping to avoid complexities during the design of any curriculum. Moreover, the ADDIE process increases the fidelity between the learning space and the performance space, which contributes to facilitating the progress of the ability of intentional learning modules [14]. Alnajdi [17] suggested that the practical interactive lessons designed in accordance with the ADDIE model engender more effective learning of and better performances by the students rather than the traditional teaching method.

In this study, we analyzed the OPC conducted during the COVID-19 pandemic in the Republic of Korea according to the ADDIE model of instructional design. First, we suitably modified the ADDIE model for OPC and included questions on design, implementation, and evaluation of the ADDIE model's components for educators and students because OPC in PE was urgently and inevitably conducted in South Korea' Universities in 2020 under COVID-19 pandemic. We measured the internal consistency of these questions using the Cronbach's $\alpha$ and evaluated the OPC performed during the COVID-19 outbreak. Through this evaluation, we intended to suggest the improvement plan for non-face-to-face classes in PE.

## 2. Materials and Methods

### 2.1. Research Design

This study was conducted at a research center from 10 March to 30 July 2020. Prior to the study, the participants received detailed briefing regarding the procedures and were subsequently asked to complete a questionnaire. It constituted 22 items regarding basic information, design, implementation, and evaluation about online PE classes according to the ADDIE design for web-based learning [18]. Among these items, the reliability of the 13 questions on design, implementation, and evaluation were analyzed and the Cronbach's $\alpha$ were 0.868. This means that the questionnaire on design, implementation,

and evaluation of online lectures on PE class was internally consistent (Table 1). In addition, the questionnaire contained the basic information including the demographic description, the type of PE (major and/or liberal arts), lecture time, preparation method and time of OPC for educators, and attending method and time of OPC for learners. Since the end of July, the data of the questionnaire sent from each university have been analyzed.

**Table 1.** Cronbach's alpha of Questions on Design, Implementation, and Evaluation.

| Items | Questionnaire | α |
|---|---|---|
| Q9 | Did you use the forums, archives and bulletin boards on online? (Have you taken classes online using the forums, archives and bulletin boards on online?) | 0.869 |
| Q10 | Did you organize and teach team projects or cooperative learning in online practical class? (Did a team project or cooperative study take place?) | 0.860 |
| Q11 | Did you teach using online media such as video editing (Did you appropriately use various techniques when you made video for online lecture) and platform (ex. Black board to upload video) in online practical class? (Was it conducted using online media such as video editing and platform?) | 0.863 |
| Q12 | Did you interact with the learners during the online practical class? (Did you interact with the professor during the online practical class?) | 0.847 |
| Q13 | Was there an advance notice for the online practical class in the context? (Was there an advance notice for the online practical class in the context?) | 0.867 |
| Q14 | Was the part needed for the online practical class supported? (Was the part needed for the online practical class supported?) | 0.856 |
| Q15 | Did any technical problems, errors, or corrections occur after the online practical class? (Did any technical problems, errors, or corrections occur after the online practical class?) | 0.883 |
| Q17 | Did you conduct the online practical class at the level of difficulty appropriate to the level of each learner? (Has the online practical class been conducted at a level suitable for the learner's individual level?) | 0.850 |
| Q18 | Did you conduct the online practical class as planned? (Was the online practical class conducted as planned?) | 0.857 |
| Q19 | Was the online practical class effectively conducted? (Was the online practical class effectively conducted?) | 0.842 |
| Q20 | Did you achieve the class goal you like to achieve through online practical class? (Did you achieve the class goal you want to achieve through online practical class?) | 0.845 |
| Q21 | Compared to face-to-face classes, has the learner's physical performance improved? (Compared to the face-to-face class, did you improve your physical skills?) | 0.857 |
| Q22 | Can you grasp the individual progress of learners after the online practical class? (Were you able to grasp your personal progress after the online practical class?) | 0.853 |
| | Cronbach's alpha | 0.868 |

Q16 is not the question consisting of 5-point Likert scale with the following variables: 1 = Not at all, 2 = No, 3 = Normal, 4 = Yes, and 5 = Very much. Therefore, it was excluded in the analysis of the internal consistency.

### 2.2. Participants

Participants who had no prior experience in OPC were recruited from the surrounding area, via word of mouth, e-mails, and SNS. In details, the semester from the beginning of March to the middle of June offered them an OPC in the domain of PE. Exclusion criteria for this study were as follows: (a) Participants who had taken part in online theoretical classes; (b) participants with sensory disorders, such as visual imperfections or hearing problems; and (c) participants with a history of brain injury or neurological disorders.

This study selected 15 universities across Korea, in consideration of regional equality, and randomly selected two professors and three students from each university. In early March 2020, the purpose of the study was announced, and at the end of the semester, the basic- and ADDIE-related questionnaires were collected. In total, 75 participants were enrolled in this study, including educators ($n = 30$) and learners ($n = 45$), who were engaged in teaching and learning practical activities, such as yoga, dance sports, swimming, and aerobic dance, at their universities, for a period of 15 weeks using the web-based delivery mechanism (online classes) during the COVID-19 pandemic crisis.

### 2.3. Modified ADDIE Model

The analysis phase of the ADDIE model is composed of two levels: Needs assessment and front-end analysis. In this phase, educators ascertain the needs of the learners [15], and the purpose of this phase is to determine the performance problems and the objectives of the instruction [18]. Thus, in this study, the need of the learners was the practical exercises, and the front-end analysis was the improvement of physical performance or skills because learners had to take the OPC in PE as their major and/or liberal arts in their Universities. During the design phase, appropriate media and instructional strategies are selected for the objectives identified during the analysis phase [15,18]. Therefore, we selected the web-based delivery (synchronous or asynchronous) as the primary delivery mode of the learning intervention, which was the best method under the COVID-19 pandemic. We also analyzed the instruction strategies for the objectives of online PE through questions, such as 'whether or not to use of various instruction methods and interaction between educator and learner'. The development phase mainly consists of the creation of course materials and instructor guides. In the development phase, the detailed interface design (preproduction), the actual web development (production), and the review and usability testing of the product (postproduction) are included [18]. In our study, the OPC in PE was urgently and inevitably conducted under COVID-19 pandemic, and thus, the educators selected the primary delivery mode in the design phase and subsequently determined the appropriate tools and the location for production or recording of online class lectures. For real-time lecture, it is very important to provide the most responsive internet experience for the educators and the learners. The educators produced the archives and the bulletin boards for the learner–content interaction and the internet forums for the learner–learner interaction. In case of any difficulties in online classes, the educators or faculties provided the technical support. In the implementation phase, educators deliver the instruction, and some support is necessary for successful implementation [15,18]. Finally, formative and summative evaluations are conducted for web-based learning interventions, just as they are for traditional classroom-based learning interventions [18]. In the evaluation phase, educators obtain feedback about their lecture, which can be appropriately addressed in the next online lecture. Thus, in this study, we investigated the characteristics of implementation and evaluation of online-based PE classes. The modified ADDIE model and the questions for online PE classes are presented in Table 2. Finally, we asked the following question in a narrative form: 'What needs to be added to make the online practical class in PE better? Feel free to describe your opinion.' Through this open-ended question, we could get loads of information about technical support, overall satisfaction in teaching-learning, and remedy for OPC in PE.

### 2.4. Data Analysis

Data were collected through an online survey. All statistical analyses were performed using the SPSS software (version 25.0, IBM Corp., Armonk, NY, USA). We used the descriptive statistical and the frequency analyses to define the basic information on subjects. We estimated the Cronbach's $\alpha$ to determine the internal consistency of this questionnaire using a 5-point Likert scale with the following variables: 1 = Not at all, 2 = No, 3 = Normal, 4 = Yes, and 5 = Very much. If the Cronbach's $\alpha$ ranged between 0.70 and 0.90, a construct was considered as being consistent [19].

**Table 2.** Modified ADDIE (Analysis, Design, Development, Implementation, Evaluation) Model and Questions for Online Physical Education Classes.

| Items | Components | Questions to Educator | | Questions to Learner | |
|---|---|---|---|---|---|
| Analysis | Needs analysis<br>Front-end analysis | - Practical exercises or activities in physical education including yoga, dance sports, aerobic dance and swimming<br>- Improvement of physical performance or skills | | | |
| Design | High-level media selection | Primary delivery mode of the learning intervention: Web-based delivery (synchronous or asynchronous) | | | |
| | | 8 | How was the online physical education class conducted? | 8 | How was the online physical education class conducted? |
| | Objective-level media selection(Instructional strategies for the objectives) | 9 | Did you use the forums, archives, and bulletin boards online? | 9 | Have you taken classes online using the forums, archives, and bulletin boards online? |
| | | 10 | Did you organize and teach team projects or cooperative learning in online practical class? | 10 | Did a team project or cooperative study take place? |
| | | 11 | Did you teach using online media such as video editing (Did you appropriately use various techniques when you made video for online lecture) and platform (e.g., Black board to upload video) in online practical class? | 11 | Was it conducted using online media such as video editing and platform? (Was the lecture video appropriately edited or Were various techniques applied to that lecture video? Could you use the black board to upload your file for interaction with lecturer?) |
| | | 12 | Did you interact with the learners during the online practical class? | 12 | Did you interact with the professor during the online practical class? |
| Development | Preproduction:<br>Production:<br>Postproduction: | · Pre-recording using appropriate tools and location after selection of the primary delivery mode in the design phase<br>· Video editing or Real-time lecture, Internet archives, Bulletin boards, and Internet forums<br>· Testing, Review, and Provision of solution for errors or problems | | | |
| Implementation | Implementation | 13 | Was there an advance notice for the online practical class in the context? | 13 | Was there an advance notice for the online practical class in the context? |
| | | 14 | Was the part needed for the online practical class supported? | 14 | Was the part needed for the online practical class supported? |
| | | 15 | Did any technical problem, error, or correction occur after the online practical class? | 15 | Did any technical problem, error, or correction occur after the online practical class? |
| Evaluation | Evaluation on learning attitude | 16 | How do you think the learner's physical exercise assignment practice was conducted? | 16 | How did you practice the physical education project? |
| | Summative | 17 | Did you conduct the online practical class at the level of difficulty appropriate to the level of each learner? | 17 | Has the online practical class been conducted at a level suitable for the learner's individual level? |
| | | 18 | Did you conduct the online practical class as planned? | 18 | Was the online practical class conducted as planned? |
| | | 19 | Was the online practical class effectively conducted? | 19 | Was the online practical class effectively conducted? |
| | | 20 | Did you achieve the class goal you want to achieve through online practical class? | 20 | Did you achieve the class goal you want to achieve through online practical class? |
| | Formative | 21 | Compared to face-to-face classes, has the learner's physical performance improved? | 21 | Compared to the face-to-face class, did you improve your physical skills? |
| | | 22 | Can you grasp the individual progress of the learners after the online practical class? | 22 | Were you able to grasp your personal progress after the online practical class? |

The analysis and development phases were already determined and were not investigated in this study using questionnaire. They were described as they were actually done.

To ascertain the normality of distribution for the examined variables, the Shapiro–Wilk test was used prior to the comparison of measurements. The Mann-Whitney U Test, a non-parametric analysis, was used to evaluate the differences in the characteristics between an educator (professor) and a learner (student). The statistical significance was determined by $p \leq 0.05$, and the data were represented as mean $\pm$ standard deviation (SD) or frequency (%), according to its scale. If there were statistically significant differences in mean between educator and learners, the *Pearson's chi-square* analysis was used. In the present study, the number of participants was 30 educators and 45 learners, thus the expected numbers may be less than 5. The result of the *chi-square* test will not be accurate if the expected numbers are not less than 5. Therefore, we performed the *chi-square* analysis after converting the five variables (1 = not at all; 2 = no; 3 = normal; 4 = yes; 5 = very much) into three categories (1 = no; 2 = normal; 3 = yes).

## 3. Results

### 3.1. Basic Information on Online PE Classes

In this study, the educators taught practical exercises in the university for 7.67 years on average, and their average age was 41.07 years. Most of the learners were freshmen (48.9%), and their average age was 21.82 years. The average online class duration for the educators and the learners was 39.50 min and 51.22 min, respectively. Most of the instructors prepared one chapter for their online lecture a week in advance, and a few performed real-time lecture. Most of the learners took the online classes at the scheduled class time, and the remaining took their online lectures on weekend or after the stipulated time (Table 3).

### 3.2. Analysis on the Design Phase of OPC in PE

During the design phase, appropriate media and instructional strategies are selected for the objectives. In this study, the primary delivery mode of the learning intervention in the practical exercise or activity education was the synchronous or asynchronous online lecture along with assignments and teaching materials. The most frequent learning method was the 'Video (asynchronous online lecture) and Assignment' and the 'Video (asynchronous online lecture), Assignment, and Teaching Material' in both the educator and the learner (Table 4).

We also investigated the instructional strategies for the objectives of the online classes in physical exercise or activity through four questions using a 5-point Likert scale with the following variables: 1 = Not at all, 2 = No, 3 = Normal, 4 = Yes, and 5 = Very much (Table 5). The mean of questions for learner–learner interaction, learner–content interaction, learner–technology interaction, and learner–instructor interaction in the OPC was not statistically different between educators and learners, and they were above average (normal). Both the educators and the learners evaluated that the online classes in practical exercise or activity did not help learner–learner interaction though team project or cooperation. They thought that a team project or cooperation was not feasible in OPC (Table 5: Q9–12).

### 3.3. Analysis on the Implementation Phase of OPC in PE

The purpose of the implementation phase is to make the web-based learning intervention available to learners [18]. Thus, some support including technical support for problems, errors, or correction must be provided. In the present study, three questions about support were provided to the educators and the learners (Table 5: Q13–15). The educators' mean about the advance notice for the OPC was significantly higher than the learners' mean. On the other hand, the educators' mean about the occurrence of technical errors or corrections after the OPC was significantly lower than the learners' mean. Therefore, we conducted the *Pearson's chi-square* analysis after converting the five-point scale into a three-point scale (Table 6).

**Table 3.** Basic Information on Online Practical Classes in Physical Education.

| Items | Educators | Learners |
|---|---|---|
| Gender | | |
| Male | 20 (66.7%) | 24 (53.3%) |
| Female | 10 (33.3%) | 21 (46.7%) |
| Age (year) | 41.07 ± 7.34 | 21.82 ± 1.86 |
| Practical class experience (year) | 7.67 ± 4.77 | - |
| Grade of students | | |
| Freshmen | - | 22 (48.9%) |
| Sophomore | - | 5 (11.1%) |
| Junior | - | 11 (24.4%) |
| Senior | - | 7 (15.6%) |
| Online practice course in physical education | | |
| Individual class | 16 (53.3%) | 21 (46.7%) |
| Group class | 14 (46.7%) | 24 (53.3%) |
| Online practical type of physical education | | |
| Major | 8 (26.7%) | 18 (40.0%) |
| Liberal arts | 9 (30.0%) | 21 (46.7%) |
| Combination (Major + Liberal) | 13 (43.3%) | 6 (13.3%) |
| Online class time (min) | 39.50 ± 20.44 | 51.22 ± 27.26 |
| Preparation method and time for Online lecture by educator | | |
| All chapters of the course are pre-recorded at once. | 3 (10.0%) | - |
| Chapters of 2 or 3 weeks are recorded at once. | 6 (20.0%) | - |
| One chapter is recorded every week. | 12 (40.0%) | - |
| It depends on the situation every week. | 7 (23.3%) | - |
| Only real-time lecture is performed on time. | 2 (6.7%) | - |
| Attending method and time for Online lecture by learner | | |
| Taking a class at the scheduled class time | - | 33 (73.3%) |
| Taking classes on weekend at once | - | 8 (17.8%) |
| Taking classes within a month (passing the scheduled class time) | - | 4 (8.9%) |
| Taking classes before test date at once | - | - |
| Rarely takes classes | - | - |

Data were presented as mean ± standard deviation (SD) in age, practical class experience, and online class duration (min). Others were presented as frequency (%).

**Table 4.** High-level Media Selection in Design: Online Teaching/Learning Method.

| Teaching/Learning Methods | Educators | Learners |
|---|---|---|
| Real-time lecture (synchronous online lecture) | 2 (6.7%) | - |
| Video (asynchronous online lecture) | 5 (16.7%) | 7 (15.6%) |
| Assignment | - | - |
| Teaching material | - | - |
| Real-time lecture + Video | 3 (10.0%) | 3 (6.7%) |
| Video + Assignment | 7 (23.3%) | 16 (35.6%) |
| Video + Teaching material | 1 (3.3%) | 1 (2.2%) |
| Real-time lecture + Video + Assignment | 2 (6.7%) | 4 (8.9%) |
| Video + Assignment + Teaching material | 8 (26.7%) | 10 (22.2%) |
| Real-time lecture + Video + Assignment + Teaching material | 2 (6.7%) | 4 (8.9%) |

This result from "Q8. How was the online physical education class conducted?". Data were presented as frequency (%).

**Table 5.** Differences in the Characteristics between the Educator and the Learner in Design, Implementation, and Evaluation.

| Item (Components) | | Question | Educators | Learners | Z (p) |
|---|---|---|---|---|---|
| Design (Online lecture's instructional strategies for the objectives) | Q9 | Use of the forums (learner-learner interaction), archives, and bulletin boards (learner-content interaction) during online lecture | 3.40 ± 1.07 | 3.27 ± 0.89 | −0.776 (0.438) |
| | Q10 | Use of a team project or cooperation in online practical class (learner-learner interaction) | 2.07 ± 0.98 | 2.53 ± 1.38 | −1.279 (0.201) |
| | Q11 | Use of online media such as video editing and platform (learner-technology interaction) | 3.60 ± 1.00 | 3.82 ± 0.91 | −0.944 (0.345) |
| | Q12 | Interaction between the professor and the students during the online practical class (learner-instructor interaction) | 3.30 ± 0.84 | 3.53 ± 0.94 | −1.163 (0.245) |
| Implementation (Learner/ instructors support in implementation) | Q13 | An advance notice for the online practical class | 4.37 ± 0.61 | 3.91 ± 0.73 | −2.666 (0.008) |
| | Q14 | The required support for the online practical class | 3.07 ± 0.94 | 3.17 ± 0.97 | −1.632 (0.103) |
| | Q15 | The occurrence of technical errors or corrections after the online practical class | 2.57 ± 0.90 | 3.33 ± 1.09 | −3.201 (0.001) |
| Evaluation (Summative: General evaluation on online lecture) | Q17 | Appropriateness in the level of difficulty (to each learner) | 3.30 ± 0.95 | 3.58 ± 0.96 | −1.211 (0.226) |
| | Q18 | Planned conduction of the online practical class | 3.37 ± 1.00 | 3.64 ± 0.96 | −1.301 (0.193) |
| | Q19 | Effectively conducted online practical class | 3.03 ± 0.93 | 3.56 ± 0.97 | −2.228 (0.026) |
| Evaluation (Formative: Effectiveness evaluation on online lecture) | Q20 | The achievement in the class goal after the online practical class | 3.20 ± 1.06 | 3.33 ±1.02 | −0.451 (0.652) |
| | Q21 | The improvement in the learners' physical performance compared to face-to-face classes | 2.13 ± 0.86 | 3.18 ± 1.19 | −3.814 (< 0.001) |
| | Q22 | The individual progress of the learners after the online practical class | 2.43 ± 1.13 | 3.49 ± 0.97 | −3.808 (<0.001) |

Data were presented as mean ± SD.

Although, the educators' mean on the provision of advance notice was significantly higher than the learners' mean, the frequency was not statistically significant between the educators and the learners. Most of the educators answered that there were fewer technical errors or corrections after the OPC. Contrarily, most of the learners thought that certain technical errors or corrections remained after the OPC. The difference in the frequency about the technical errors between the educators and the learners was statistically significant (Table 6: Q13, 15).

*3.4. Analysis on the Evaluation Phase of OPC in PE*

After implementing the instruction, we evaluated whether the online lectures on practical physical exercises achieved their intended goal and what attitudes were shown by learners from the perspective of the educator and the learner. Most of the educators thought that the students were not enthusiastically involved in the online classes and that most of the students practiced their physical exercises only for submission of assignment, not for improvement. Even the learners felt that they performed the physical exercises merely for submitting assignments during the semester. In fact, some of the educators thought that the students rarely practiced the physical exercises that they learned from the OPC. On the other hand, the learners opined that they practiced the physical exercise assignments although they did not do them enthusiastically (Table 7).



**Table 6.** The *Pearson's Chi-square* Analysis after Converting the Five-point Scale into a Three-point Scale.

|  |  |  | No | Normal | Yes | Total | $\chi^2$ (*p*) |
|---|---|---|---|---|---|---|---|
| Q13 | Educators | Frequency | 0 | 2 | 28 | 30 | 4.834 |
|  |  | (% of group) | (0%) | (6.7%) | (93.3%) | (100.0%) | (0.089) |
|  | Learners | Frequency | 1 | 11 | 33 | 45 |  |
|  |  | (% of group) | (2.2%) | (24.4%) | (73.3%) | (100.0%) |  |
|  |  | Total | 1 | 13 | 61 | 75 |  |
|  |  | (%) | (1.3%) | (17.3%) | (81.3%) | (100.0%) |  |
| Q15 | Educators | Frequency | 16 | 10 | 4 | 30 | 12.434 |
|  |  | (% of group) | (53.3%) | (33.3%) | (13.3%) | (100.0%) | (0.002) |
|  | Learners | Frequency | 10 | 12 | 23 | 45 |  |
|  |  | (% of group) | (22.2%) | (26.7%) | (51.1%) | (100.0%) |  |
|  |  | Total | 26 | 22 | 27 | 75 |  |
|  |  | (%) | (34.7%) | (29.3%) | (36.0%) | (100.0%) |  |
| Q19 | Educators | Frequency | 7 | 14 | 9 | 30 | 5.921 |
|  |  | (% of group) | (23.3%) | (46.7%) | (30.0%) | (100.0%) | (0.052) |
|  | Learners | Frequency | 8 | 11 | 26 | 45 |  |
|  |  | (% of group) | (17.8%) | (24.4%) | (57.8%) | (100.0%) |  |
|  |  | Total | 15 | 25 | 35 | 75 |  |
|  |  | (%) | (20.0%) | (33.3%) | (46.7%) | (100.0%) |  |
| Q21 | Educators | Frequency | 23 | 4 | 3 | 30 | 18.384 |
|  |  | (% of group) | (76.7%) | (13.3%) | (10.0%) | (100.0%) | (0.001) |
|  | Learners | Frequency | 12 | 14 | 19 | 45 |  |
|  |  | (% of group) | (26.7%) | (31.1%) | (42.2%) | (100.0%) |  |
|  |  | Total | 35 | 18 | 22 | 75 |  |
|  |  | (%) | (46.7%) | (24.0%) | (29.3%) | (100.0%) |  |
| Q22 | Educators | Frequency | 17 | 7 | 6 | 30 | 16.219 |
|  |  | (% of group) | (56.7%) | (23.3%) | (20.0%) | (100.0%) | (0.001) |
|  | Learners | Frequency | 6 | 17 | 22 | 45 |  |
|  |  | (% of group) | (13.3%) | (37.8%) | (48.9%) | (100.0%) |  |
|  |  | Total | 23 | 24 | 28 | 75 |  |
|  |  | (%) | (30.7%) | (32.0%) | (37.3%) | (100.0%) |  |

Data were presented as frequency and percentage of group.

**Table 7.** Learning Attitude of Physical Exercise Assignment Measured by Professor and Student.

| Items | Educators | Learners |
|---|---|---|
|  | Frequency (%) | Frequency (%) |
| Students (or I) keep practiced physical exercise assignment every day for self-advancement. | 3 (10.0%) | 6 (13.3%) |
| Students (or I) practice physical exercise assignment, but do not perform it enthusiastically. | 5 (16.7%) | 15 (33.3%) |
| Students (or I) practice physical exercise only for submission of assignment. | 16 (53.3%) | 21 (46.7%) |
| Students (or I) seldom practice physical exercise. | 6 (20.0%) | 3 (6.7%) |

This result from "Q 16. How do you think the learner's physical exercise assignment practice was conducted? (for educators) or "Q16. How did you practice the physical education project? (for learners)".

We also conducted the summative evaluation—the appropriateness in the level of difficulty, the performance degree, and the effectiveness of conduction (Table 5: Q17–19). The average scores in the appropriateness in the level of difficulty and in the performance degree were similar between the educators and the learners. Both thought that the level of difficulty was appropriate, and the class was taught as planned to some degree (above normal). However, the mean of effectiveness in the conduction of OPC was significantly different between them, and the educators' mean was significantly lower than that of the learners. Thus, we performed the *chi-square* analysis, but the distribution of frequency was not statistically different between the educators and the learners. However, 46.7% of the

educators thought that the effectiveness in the conduction of OPC was normal, but 57.8% of the learners felt that the OPC were effectively conducted (Table 6: Q17).

Finally, we evaluated the achievement of the class objective, the improvement in the learner's physical performance, and the individual progress. There were statistically significant differences in the scores on the improvement and the individual progress of learners (Table 5: Q20–22). Therefore, we analyzed the distribution of frequency, and the results showed that 76.7% of the educators thought that the learners' physical performance did not improve compared to the face-to-face classes. They also evaluated that they could not determine the learners' improvement or advancement after the OPC. However, 42.2% of the learners believed that their physical performance improved, and 48.9% of the learners felt that their physical performance improved after the OPC (Table 6: Q21, 22).

## 4. Discussion

In the present study, the practical exercises, or activities, including yoga, dance sports, swimming, and aerobic dance, were delivered by web-based modes: Video (asynchronous online lecture) or real-time lecture (synchronous online lecture). A synchronous online lecture means that all the learners and the educators are logged in at the same time, and that all of them communicate directly and virtually with one another. In an asynchronous online lecture, the learners take the prepared or pre-designed lesson, which is often available 24 h per day and 7 days a week. In our study, two educators (6.7%) preferred the real-time lecture to ensure that the students accurately followed the movement or posture. Most of the educators combined their online classes with assignments or teaching materials. Similarly, the learners attended their practical classes through online mode and worked on the assignments and teaching material (Table 4). Many educators prepared their OPC every week or 2–3 weeks before the scheduled online class. Some educators reported that they prepared their OPC depending on the situation. On the other hand, most of the learners took their OPC at the stipulated class time (Table 1). According to the ADDIE model, we surveyed 13 questions about the design, implementation, and evaluation of the OPC in PE through a questionnaire, the construct of which was considered as being consistent (Cronbach's $\alpha$ = 0.868) (Table 3).

Lee et al. [18] suggested that accommodating interactions between learner–learner, learner–content, learner–instructor, and learner–technology in the web-based learning environment mediates the physical distance between the learners and the instructor, and between individual learners. Therefore, these four interactions should be considered in the 'design phase'. In the present study, we surveyed these four interactions. Both the educators and the learners recorded that forums for learner–learner interaction, the bulletin boards for learner–content interaction, and the video and platform for learner–technology interaction were used better than average (normal) in the OPC. They also felt that the interaction between the educators and the learners during the OPC was above average (normal) (Table 5: Q9, 11, 12). Previous studies reported that there was more interaction between teachers and students in an online environment [20], and that students were good at online communication [21]. For example, Rabe-Hemp et al. [22] suggested that the asynchronous nature of the online course allowed the student and the professor a more thoughtful process of communication.

However, as expected, the team project or cooperation was not effective in the OPC (Table 5: Q10). Cooperative learning is defined as 'students learning with, by and for each other' [23] and is reported to help PE achieve the four basic learning outcomes: Cognitive (increasing students' knowledge on related strategies and skills), social (fostering good social relations), physical (augmenting levels of physical activity through the increase of motor skills, sport techniques, and performance), and affective (self-confidence, self-esteem, and motivation) [24,25]. Many studies reported that face-to-face interaction and group processing, two elements of cooperative learning, offered increased opportunities for student interaction, leading to academic learning [26,27]. In the present study, most of the learners took their classes only at the scheduled time, and the mean of class duration was 51.22 min



in this study. Although they used the online forums for learner–learner interaction, the use of the forum was inferred to be inadequate for increasing the opportunities for student interaction. Furthermore, group processing or team project was impossible in the OPC.

Some support is reportedly necessary for educators and learners in the 'implementation phase'. For example, a learner needs certain support, such as technical support for problems, answers to questions about content or a particular technique, content support for problems, and administrative support for registration, testing, and certification. Similarly, educators need technical support [18]. In the present study, both the educators and the learners thought that a certain support was needed for the OPC. Most of the educators thought that their OPC were well prepared, and thus there were few technical errors during or after an online class. On the other hand, most of the learners reported that there were a few technical errors after the online classes although there was an advance notice for such classes (Tables 5 and 6: Q15). In the narrative form question, some of the educators and learners answered that technical support for video editing, filming, and sound technology were needed, and that the faculty or the university must provide the tools of practical classes to both the professors and the students. A certain educator thought that it was necessary to support a space optimized for real-time video lectures (e.g., a classroom equipped with a computer, large monitor, wireless microphone, and sound equipment). Moreover, it was felt that an adequate technical support staff is imperative for all web-based learning initiatives, and that this staff needs to be trained and available to assist both the students and the instructors. In addition, this technical support should be provided by the faulty or the university round the clock [18]. However, in the present study, these OPC in PE were suddenly implemented in Korea because of the COVID-19 pandemic, and thus every university might not be equipped with adequate well-trained and technical staff. The instructors also need ongoing maintenance support to help them with the changes that may be required in the course content, and they need training on teaching in a web-based environment prior to their first online teaching experience.

In the present study, we evaluated the appropriateness in the level of difficulty, performance degree, and the effectiveness of conduction. Both the educators and the learners thought that the level of difficulty was appropriate, that the classes were taught as planned to some degree (above normal), and that the OPC was effectively conducted. Both also felt that the students could achieve their goal through the OPC (Table 5: Q17–20). However, they differently evaluated the improvement in the learners' physical performance and the individual progress of learners. Most of the educators thought that the learners' physical performance was not improved compared to face-to-face classes, and they could not determine the learners' improvement or advancement after the OPC. On the other hand, most of the learners (42.2%) were of the impression that their physical performance was improved when it was compared to face-to-face class, and 48.9% of the learners felt that they were improved after OPC (Tables 5 and 6: Q21, 22). It should thus be noted that the difference between educators and learners in terms of improvement at the end of the semester is a result of their difference in ability to grasp practical skills. In other words, instructors who have been mainly teaching PE in off-line classes or face-to-face classes are not satisfied, no matter how well the learners have performed, whereas learners have the satisfaction of 'learning new things' for the first few practical lessons.

Generally, many previous studies reported that distance education courses were as effective as face-to-face courses [28–30]. These studies reported that student achievement in distance education course could be as good as that in face-to-face education; some studies even suggested that achievement in distance education was higher than that in face-to-face education [29,31]. In the present study, most of the students evaluated that they had improved to above average, and that they could grasp their own advancement. However, most of the educators thought otherwise. Through narrative form questions by educators, we found that students appeared to follow the posture or movement well when viewed through the video. Thus, educators thought that students understood the OPC well. However, they detected the students' wrong motion or posture at the final evaluation.

In the traditional classroom, the importance of the interaction between the instructor and the students has been considered as essential [32], and this interaction often occurs in 'the form of feedback' [33]. In the online class, providing feedback may be a challenging task for instructors, especially those who have spent most of their teaching careers in the traditional face-to-face environment [34]. The physical educators have taught the practical exercise or activity in face-to-face environment and have directly provided the immediate feedback demonstrating or correcting the posture or movement on site. Through timely and quality feedback, students can improve their sports skills. However, there are some limitations for them to provide feedback in the OPC. In the open-ended question of this study, many educators indicated the issue of 'insufficient feedback' or 'difficulty in providing timely and quality feedback' in the online class format. In the class, educators are mediators between the content (or practical training) and the learners. Especially, correct posture and movement are required in physical exercise or activity. Therefore, educators identify the students' individual levels and directly correct their posture and movement in a face-to-face class. However, they cannot do so in an online class, resulting in degraded quality of the classes. Some students stated that they were not sure whether they were performing the physical exercises properly, and that they were disappointed that they did not have the opportunity to receive the appropriate feedback.

The practical exercise or activity in PE is a kind of active learning in which students learn by 'doing' [35,36]. Austin and Mescia [37] suggested that through this process students engage in higher-order thinking, including analysis, synthesis, and evaluation, and this allows them to assimilate, apply and, retain newly acquired information. Thus, the online instructors must make special efforts regarding timely and quality feedback. In the present study, many educators mentioned the importance of feedback appropriate to the individual level in PE and suggested that physical educators must provide the feedback obligatorily in OPC. Specifically, it would be good to develop various videos so that the students can practice and manage themselves according to their individual level; alternatively, the detailed feedback can be provided through an intermediate check: For example, students can take a video of a posture or movement using a smartphone after the online class, and then they can submit the video to the educator. The online instructor can assess the individual posture and provide the personalized feedback to the students. Healy et al. [33] also suggested that students may work in small groups to devise inclusive PE lesson plans within online course. This collaborative work or collective group allows students to interact with each other, and the online instructor can provide more detailed feedback to the students in the collective group [33]. This collective group may be beneficial to the cooperative learning method, which was evaluated as weak in the present study (Table 5: Q10).

In the current study, we measured the online practical activity classes performed during the COVID-19 pandemic, from the perspective of educators and learners. Overall, they similarly evaluated the OPC in PE, except the improvement in the learner's physical performance and the individual progress of learners after the online class. Many physical educators emphasized the importance of timely and quality feedback in the OPC mode, and thus suggested the mandatory provision of feedback for the successful execution of OPC in PE. Based on the results of this study, we suggest some prerequisites which must be met in order to provide timely and quality feedback in the OPC in PE. First, the university should provide sufficient time and technical support, including video editing, filming, and sound technology, for the physical educators. The physical instructors need time to create the active learning strategies, which comprise providing feedback to the learners in online curriculum. Specifically, the technical support is critical to all the phases of planning, implementing, and evaluating the OPC; for example, the educators must design their educational materials including videos so that students can practice and manage them according to their individual level. In addition, these videos should be presented to the students without any technical errors. In addition, the university should provide adequate technical staff to the educators and the learners. Second, instructors or educators need to

be trained in the web-based environment prior to their online teaching to keep pace with the evolving education environment. Finally, educators must devise ways that students can voluntarily engage in the class through the cooperative work or team project. They also need to devise a fair and accurate way to evaluate the learners' performance in OPC.

## 5. Conclusions

This study confirmed whether the urgently performed online practical classes were conducted in a suitable form according to the ADDIE model from the perspective of educators and learners. From the results of this study, we found that timely and quality feedback should be provided for the successful online practical classes and this study provided some suggestions for the successful execution of online practical classes in physical education: (1) The university should provide sufficient time and technical support; (2) educators need to be trained on web-based environment prior to their online teaching; (3) educators have to devise ways to encourage students' involvement and fair ways to evaluate. Although it is not easy to conduct online practical classes, we can step closer to conducting such classes for physical education through timely and quality feedback. This study is worthy as it provides some suggestions for the successful execution of online practical classes in physical education. However, this study has some limitations. Firstly, the number of participants was small. Secondly, the types of questions were not sufficient and their validation was not tested, although we estimated the Cronbach's $\alpha$ to determine the internal consistency of the questionnaire. Thus, more comprehensive research is needed with more participants and through more in-depth questionnaires.

**Author Contributions:** Conceptualization, Y.J. and J.Y.; methodology, Y.J.; software, J.Y.; validation, Y.J. and J.Y.; formal analysis, Y.J.; investigation, Y.J.; resources, J.Y.; data curation, J.Y.; writing—original draft preparation, Y.J.; writing—review and editing, Y.J.; visualization, J.Y.; supervision, Y.J.; project administration, J.Y. All authors have read and agreed to the published version of the manuscript.

**Funding:** This research received no external funding.

**Conflicts of Interest:** The authors declare no conflict of interest.

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
