# Peer review of "Analysis of Online Classes in Physical Education during the COVID-19 Pandemic"

_education, doi:10.3390/educsci11010003_

Round 1
Reviewer 1 Report
The proposal is correct in general but we have some suggestions:
- The theorical and experimental background is scarce and should be completed with theoretical models and previous experiences developed at an international level.
- In general terms the design research is appropriate but it could be debatable whether self-perception instruments are sufficient to assess the effectiveness of a teaching model.
- I suggested to explain some information about the general context of research: activities, programs, methodology....
- The validation process of the instrument used does not appear; was created or is an adaptation of an existing one? You only explain the reliability.
- The general data analysis seems correct
- The results are not very clear. There are some parts that could move to other part of the paper. For example, table 1 present the results about the profile and characteristics of the classroom PE: table 3 could move to the experimental design part in order to explain the instrument better.
- Perhaps it could be to define better what the authors want to evaluate at the front-end analysis: it is not the same the physical performance, skills or attitudes. These are concepts that appears in different parts of the proposal.
- Some parts of the discussion seems not to be supported by the results. It seems to be an interpretation. For example: “In other words, instructors who have been teaching PE for a long time are not satisfied, no matter how well the learners have performed, whereas learners have the satisfaction of 'learning new things' for the first few practical lessons”
- In the conclusions there is a reference to the Olympic athletes that I don’t understand the meaning.
Author Response
Answers to 1st reviewer’s comments
Comments and Suggestions for Authors
The proposal is correct in general but we have some suggestions:
- The theorical and experimental background is scarce and should be completed with theoretical models and previous experiences developed at an international level.
Answer: According to reviewer’s comment, we reinforced the introductory background by citing previous studies on the theorical and experimental background, and we added more references. Revised introduction is as follows and we represented the specific modifications in response to the comments by blue-letters in our manuscript (1. Introduction).
For example,
- Introduction
…
With the growth of online learning or instruction, a pedagogically effective instructional design model for online education is strongly required to facilitate the development and delivery of engaging online learning environments because a poorly designed online lecture makes learners get lost, loses their interest and feel distressed [35]. Instructional design or instructional systematic design is defined as “the practice of creating instructional experiences which make the acquisition of knowledge and skill more efficient, effective, and appealing”, and an effective instructional design is very beneficial for instructors and learners [35, 36]. The ADDIE model is one of frequently cited instructional design models which is utilized to create training programs in a steady and solid form in diverse fields [12-14]. The acronym ADDIE denotes the following five phases: analysis, design, development, implementation, and evaluation [15]. The analysis phase in which educators ascertain the needs of the learners involves setting educational goals and determining what needs to be taught. During analysis phase, educators can use various methods including surveys, exert consensus, and opinions of graduates of the program to gather the information. In design phase, educators describe how to deliver their instruction to meet the objectives under a broad overview or blueprint. The design phase includes selecting the optimal methods and creating useful learning goals. The component of the instruction is concretely planned in the development phase, and then educators deliver the instruction in the implementation. Finally, in the evaluation phase, educators get feedback on the instruction or lecture and appropriately adjust it to their program of instruction. One of important methods evaluating the educational effectiveness is to objectively measure how well the learners have acquired the knowledge, skills or attitudes being taught [13, 34].
…
[Added References]
- 34. Sanaiey, N.Z.; Karamnejad, S.; Rezaee, R. Educational needs of family physicians in the domains of health and conformity with continuing education in Fasa University of Medical Sciences. J Adv Med Educ Prof. 2015, 3, 84-89. PMID: 25927073, PMCID: PMC4403570
- Chen, L. A model for effective online instructional design. Literacy Information and Computer Education Journal (LICEJ) 2016, 6, 2303-2308. https://www.researchgate.net/profile/Li_Ling_Chen2/publication/326865522_A_Model_for_Effective_Online_Instructional_Design/links/5bd4f18492851c6b27931831/A-Model-for-Effective-Online-Instructional-Design.pdf
- Roblyer, M.D. Introduction to systematic instructional design for traditional, online, and blended environments. New Jersey: Pearson Eudcation, Inc. 2015.
I wish this response can satisfy you, and appreciate your comment. Owing to your comment, I think that our study will be more improved.
- In general terms the design research is appropriate but it could be debatable whether self-perception instruments are sufficient to assess the effectiveness of a teaching model.
Answer: As you know, in the present study, we analyzed the online practical classes in physical education during the COVID-19 pandemic in Korea according to the ADDIE model of instructional design, and we evaluated and compared the opinions (or responses) on the online practical classes using self-perception questionnaires. As you know, there were some degrees and/or disagrees between them on the same questions. For example, both the educators and the learners thought that a certain support was needed for the OPC. Most of the educators thought that their OPC were well prepared, and thus there were few technical errors during or after an online class. On the other hand, most of the learners reported that there were a few technical errors after the online classes although there was an advance notice for such classes (Table 5, 6: Q15).
Like this, educators and learners had different or similar opinions or evaluations on the same issue, and we described them in the section of results and discussed them. This is consistent with the intent of our study. Moreover, the effectiveness of the online practical classes was evaluated as the achievement in the class goal, the improvement in the learners’ physical performance, and the individual progress of the learners. However, in the present study, our participants taught or learned various practical activities such as yoga, dance sports, aerobic dance and swimming in fifteen universities of Korea. Thus, we could not evaluate their achievement or improvement using the fixed or typical instrument. We had no choice but to use self-perception evaluation.
I hope this response can satisfy you and you understand what we have used this self-perception instrument to evaluate the effectiveness of teaching model.
- I suggested to explain some information about the general context of research: activities, programs, methodology....
Answer: As I mentioned above, participants of this study were educators and learners who were engaged in teaching and learning practical activities, such as yoga, dance sports, aerobic and swimming. And, we described it in our manuscript (the section of ‘Participants of Materials and Methods’ and Discussion).
For example,
Participants
…
This study selected fifteen universities across Korea, in consideration of regional equality, and randomly selected two professors and three students from each university. In early March, 2020, the purpose of the study was announced, and at the end of the semester, the basic- and ADDIE-related questionnaires were collected. In total 75 participants were enrolled in this study, including educators (n = 30) and learners (n = 45), who were engaged in teaching and learning practical activities, such as yoga, dance sports, swimming, and aerobic dance, at their universities, for a period of 15 weeks using the web-based delivery mechanism (online classes) during the COVID-19 pandemic crisis. Table 3 presents the demographic and basic information on OPC.
- Discussion
In the present study, the practical exercises, or activities, including yoga, dance sports, swimming, and aerobic dance, were delivered by web-based modes: video (asynchronous online lecture) or real-time lecture (synchronous online lecture).
…
I wish this response can satisfy you, and appreciate your comment.
- The validation process of the instrument used does not appear; was created or is an adaptation of an existing one? You only explain the reliability.
Answer: In the present study, we tested the reliability by estimating the Cronbach’s alpha, but we did not test the validation. Therefore, we described this in Conclusion as the limitation of our study with revision of Conclusion according to your last comment.
We represented the specific modifications in response to the comments by blue-letters in our manuscript.
For example,
- Conclusions
This study confirmed whether the urgently performed online practical classes were conducted in a suitable form according to the ADDIE model from the perspective of educators and learners. From the results of this study, we found that timely and quality feedback should be provided for the successful online practical classes and this study provided some suggestions for the successful execution of online practical classes in physical education: (1) the university should provide sufficient time and technical support; (2) educators need to be trained on web-based environment prior to their online teaching; (3) educators have to devise ways to encourage students’ involvement and fair ways to evaluate. Although it is not easy to conduct online practical classes, we can step closer to conducting such classes for physical education through timely and quality feedback. This study is worthy as it provides some suggestions for the successful execution of online practical classes in physical education. However, this study has some limitations. Firstly, the number of participants was small. Secondly, the types of questions were not sufficient and their validation was not tested, although we estimated the Cronbach’s α to determine the internal consistency of the questionnaire. Thus, more comprehensive research is needed with more participants and through more in-depth questionnaires.
I wish this response can satisfy you, and appreciate your comment. Owing to your comment, I think that our study will be more improved.
- The general data analysis seems correct
Answer: I appreciate your comment.
- The results are not very clear. There are some parts that could move to other part of the paper. For example, table 1 present the results about the profile and characteristics of the classroom PE: table 3 could move to the experimental design part in order to explain the instrument better.
Answer: As reviewer pointed out, table 1 contained the results on the profile and characteristics of the physical education. Originally, we had separated these results into several tables. However, according to the submission format of the Journal, we had no choice but to combine them. However, according to reviewer’s comment, we moved table 1 into the results and changed the table number. And, we brought table 3 to the experimental design part, and changed the table number.
For example (The representation of Table 1 and 3 is skipped.),
- Materials and methods
Experimental Design
This study was conducted at a research center from March 10 to July 30, 2020. Prior to the study, the participants received detailed briefing regarding the procedures and were subsequently asked to complete a questionnaire. It constituted 22 items regarding basic information, design, implementation, and evaluation about online PE classes according to the ADDIE design for web-based learning [16]. Among tese items, the reliability of the thirteen questions on design, implementation, and evaluation was analyzed and the Cronbach’s α was 0.868. This means that the questionnaire on design, implementation, and evaluation of online lectures on PE class was internally consistent (Table 1). Since the end of July, the data of the questionnaire sent from each university has been analyzed.
Table 1. Cronbach’s alpha of questions on design, implementation and evaluation.
…
- Results
Basic information on online PE classes
In this study, the educators taught practical exercises in the university for 7.67 years on average, and their average age was 41.07 years. Most of the learners were freshmen (48.9 %), and their average age was 21.82 years. The average online class duration for the educators and the learners was 39.50 min and 51.22 min, respectively. Most of the instructors prepared one chapter for their online lecture a week in advance, and a few performed real-time lecture. Most of the learners took the online classes at the scheduled class time, and the remaining took their online lectures on weekend or after the stipulated time (Table 3).
Table 3. Basic Information on Online Practical Classes in Physical Education.
I wish this response can satisfy you, and appreciate your comment. Owing to your comment, I think that our study will be more improved.
- Perhaps it could be to define better what the authors want to evaluate at the front-end analysis: it is not the same the physical performance, skills or attitudes. These are concepts that appears in different parts of the proposal.
Answer: The front-end analysis (FEA) is the “blueprint” for creating instruction. The FEA defines the requirement, describes the ideal performance or instruction to meet the requirements, and identifies acceptable alternatives. Typically, FEA is used to define the current and desired performance states, and identify the performance gap between the two. In the present study, we evaluated and compared the opinions (or responses) on the online practical classes using self-perception questionnaires. Thus, as reviewer pointed out, it may be more desirable to use the FEA in this study. However, in the present study, we intended to analyze the online practical classes conducted during COVID-19 pandemic according to the ADDIE model because it is not easy to conduct the online practical classes in physical education, and they were urgently and inevitably conducted under COVID-19 pandemic.
The instructional design is defined as “systematic and reflective process of translating principles of learning and instruction into plans of instructional materials, activities, information sources and evaluation”, and this process of design usually consist of five key components: Analysis, Design, Development, Implementation and Evaluation (Smith and Ragan, 1999). Weston (1989) stated that FEA is very important for developing country instructional technology interventions. However, this type of analysis is not designed to analyze the broader context (Arias and Clark, 2007).
I hope this response can satisfy you and you understand what we have used the ADDIE model in the present study.
[References]
Smith, P. L., & Ragan, T. J. (1999). Instructional Design (2nd Ed.). New York: Wiley
Weston, C. B. (1989). Critical Factors for Educational Technology Interventions in Developing Countries. Educational & Training Technology International, 26(2), 122-128.
Arias, S., & Clark, K.A. (2007). Instructional technology in developing countries: A contextual analysis approach. Tech Trends, 48(4), 52-70.
- Some parts of the discussion seems not to be supported by the results. It seems to be an interpretation. For example: “In other words, instructors who have been teaching PE for a long time are not satisfied, no matter how well the learners have performed, whereas learners have the satisfaction of 'learning new things' for the first few practical lessons”
Answer: In the section of ‘Measures of Materials and Methods’, we described “… Finally, we asked the following question in a narrative form: ‘What needs to be added to make the online practical class in PE better? Feel free to describe your opinion.’ Through this open-ended question, we could get loads of information about technical support, overall satisfaction in teaching-learning, and remedy for OPC in PE.”. In the present study, we mixed multiple choice questions and narrative form questions, and we described the responses from the narrative form questions. Thus, according to reviewer’ comment, we described the results from the multiple choice questions and narrative form questions using the phrase ‘In narrative form question’ and other expressions, and we revised the Discussion.
For example,
- Discussion
… In narrative form question, some of the educators and learners answered that technical support for video editing, filming, and sound technology were needed, and that the faculty or the university must provide the tools of practical classes to both the professors and the students. A certain educator thought that it was necessary to support a space optimized for real-time video lectures (e.g. a classroom equipped with a computer, large monitor, wireless microphone, and sound equipment).
… It should thus be noted that the difference between educators and learners in terms of improvement at the end of the semester is a result of their difference in ability to grasp practical skills. In other words, instructors who have been mainly teaching PE in off-line classes or face-to-face classes are not satisfied, no matter how well the learners have performed, whereas learners have the satisfaction of 'learning new things' for the first few practical lessons.
We represented the specific modifications in response to the comments by blue-letters, and we wish this response can satisfy you, and appreciate your comment. Owing to your comment, I think that our study will be more improved.
- In the conclusions there is a reference to the Olympic athletes that I don’t understand the meaning.
Answer: According to reviewer’s comment, we deleted the sentence about the Olympic athletes, and revised Conclusion. Revised Conclusion was represented the answer to your upper other comment on the Validation of instrument. I appreciate your comment.
Re-submission Date: 09 December 2020

Reviewer 2 Report
This work is actual and useful. The introductory background should be improved and more references should be added. The research method should be better specified and it should have a better order. PArticipants should not be the first section in this part. The conclusions should be clearer. Such a discussion section deserve a better way of concliude the conclussions section. They are not clear and enough.
Author Response
Answers to 2nd reviewer’s comments
Comments and Suggestions for Authors
- This work is actual and useful.
Answer: I appreciate your comment.
- The introductory background should be improved and more references should be added.
Answer: According to reviewer’s comment, we reinforced the introductory background by citing previous studies, and we added more references. Revised introduction is as follows and we represented the specific modifications in response to the comments by blue-letters in our manuscript (Introduction).
For example,
- Introduction
…
With the growth of online learning or instruction, a pedagogically effective instructional design model for online education is strongly required to facilitate the development and delivery of engaging online learning environments because a poorly designed online lecture makes learners get lost, loses their interest and feel distressed [35]. Instructional design or instructional systematic design is defined as “the practice of creating instructional experiences which make the acquisition of knowledge and skill more efficient, effective, and appealing”, and an effective instructional design is very beneficial for instructors and learners [35, 36]. The ADDIE model is one of frequently cited instructional design models which is utilized to create training programs in a steady and solid form in diverse fields [12-14]. The acronym ADDIE denotes the following five phases: analysis, design, development, implementation, and evaluation [15]. The analysis phase in which educators ascertain the needs of the learners involves setting educational goals and determining what needs to be taught. During analysis phase, educators can use various methods including surveys, exert consensus, and opinions of graduates of the program to gather the information. In design phase, educators describe how to deliver their instruction to meet the objectives under a broad overview or blueprint. The design phase includes selecting the optimal methods and creating useful learning goals. The component of the instruction is concretely planned in the development phase, and then educators deliver the instruction in the implementation. Finally, in the evaluation phase, educators get feedback on the instruction or lecture and appropriately adjust it to their program of instruction. One of important methods evaluating the educational effectiveness is to objectively measure how well the learners have acquired the knowledge, skills or attitudes being taught [13, 34].
…
[Added References]
- 34. Sanaiey, N.Z.; Karamnejad, S.; Rezaee, R. Educational needs of family physicians in the domains of health and conformity with continuing education in Fasa University of Medical Sciences. J Adv Med Educ Prof. 2015, 3, 84-89. PMID: 25927073, PMCID: PMC4403570
- Chen, L. A model for effective online instructional design. Literacy Information and Computer Education Journal (LICEJ) 2016, 6, 2303-2308. https://www.researchgate.net/profile/Li_Ling_Chen2/publication/326865522_A_Model_for_Effective_Online_Instructional_Design/links/5bd4f18492851c6b27931831/A-Model-for-Effective-Online-Instructional-Design.pdf
- Roblyer, M.D. Introduction to systematic instructional design for traditional, online, and blended environments. New Jersey: Pearson Eudcation, Inc. 2015.
I wish this response can satisfy you, and appreciate your comment. Owing to your comment, I think that our study will be more improved.
- The research method should be better specified and it should have a better order. PArticipants should not be the first section in this part.
Answer: According to reviewer’s comment, we changed the order of Materials and methods. We brought ‘Experimental Design’ in front of ‘Participant’. And, we re-arranged the order of Table #1 and #3, and changed the number of table.
For example (The representation of Table 1 and 3 is skipped.),
- Materials and methods
Experimental Design
This study was conducted at a research center from March 10 to July 30, 2020. Prior to the study, the participants received detailed briefing regarding the procedures and were subsequently asked to complete a questionnaire. It constituted 22 items regarding basic information, design, implementation, and evaluation about online PE classes according to the ADDIE design for web-based learning [16]. Among tese items, the reliability of the thirteen questions on design, implementation, and evaluation was analyzed and the Cronbach’s α was 0.868. This means that the questionnaire on design, implementation, and evaluation of online lectures on PE class was internally consistent (Table 1). Since the end of July, the data of the questionnaire sent from each university has been analyzed.
Table 1. Cronbach’s alpha of questions on design, implementation and evaluation.
…
- Results
Basic information on online PE classes
In this study, the educators taught practical exercises in the university for 7.67 years on average, and their average age was 41.07 years. Most of the learners were freshmen (48.9 %), and their average age was 21.82 years. The average online class duration for the educators and the learners was 39.50 min and 51.22 min, respectively. Most of the instructors prepared one chapter for their online lecture a week in advance, and a few performed real-time lecture. Most of the learners took the online classes at the scheduled class time, and the remaining took their online lectures on weekend or after the stipulated time (Table 3).
Table 3. Basic Information on Online Practical Classes in Physical Education.
I wish this response can satisfy you, and appreciate your comment. Owing to your comment, I think that our study will be more improved.
- The conclusions should be clearer. Such a discussion section deserve a better way of concliude the conclussions section. They are not clear and enough.
Answer: According to reviewer’s comment, we revised the Conclusion. Revised conclusion was as follows;
- Conclusions
This study confirmed whether the urgently performed online practical classes were conducted in a suitable form according to the ADDIE model from the perspective of educators and learners. From the results of this study, we found that timely and quality feedback should be provided for the successful online practical classes and this study provided some suggestions for the successful execution of online practical classes in physical education: (1) the university should provide sufficient time and technical support; (2) educators need to be trained on web-based environment prior to their online teaching; (3) educators have to devise ways to encourage students’ involvement and fair ways to evaluate. Although it is not easy to conduct online practical classes, we can step closer to conducting such classes for physical education through timely and quality feedback. This study is worthy as it provides some suggestions for the successful execution of online practical classes in physical education. However, this study has some limitations. Firstly, the number of participants was small. Secondly, the types of questions were not sufficient and their validation was not tested, although we estimated the Cronbach’s α to determine the internal consistency of the questionnaire. Thus, more comprehensive research is needed with more participants and through more in-depth questionnaires.
I wish this response can satisfy you, and appreciate your comment. Owing to your comment, I think that our study will be more improved.
Re-submission Date: 09 December 2020

Round 2
Reviewer 2 Report
The work has been improved, but I still think that the method section should be better specified.
Author Response
Answers to 2nd reviewer’s comments
Comments and Suggestions for Authors
Comment: The work has been improved, but I still think that the method section should be better specified.
Answer: According to your 1st comment about the research method, we revised our manuscript (the section of ‘Materials and methods’) using re-arrangement of the order. However, you suggested that the method section should be better specified.
In the present study, we analysed the online practice class conducted during the COVID-19 pandemic according to the ADDIE model of instructional design. However, among ADDIE, the analysis phase and development phase were already determined according to following reasons:
- In the analysis phase of the ADDIE model is consisted of two levels: needs assessment and front-end analysis. And, educators ascertain the needs of the learners and the purpose of this phase. However, the needs and purpose of the practical classes were already determined in this study because learners had to take their practical classes in the physical education as their major and/or liberal arts in their Universities. Therefore, the analysis phase had already been determined in this study, and the analysis phase of ADDIE model was not investigated in this study.
- As you know, the development phase is composed of the creation of course materials and instructor guides. Originally, in the development phase, the detailed interface design, the actual web development and the review and usability testing of the product are included. However, in South Korea, the practical classes in the physical education were urgently and inevitably conducted under COVID-19 pandemic. Thus, educators had no choice but to pre-recording using appropriate tools and location after selection of the primary delivery mode in the design phase (Preproduction), video editing or real-time lecture with various materials (internet archives and bulletin boards) (Production), and testing, review and provision of solutions for errors with the faculty or University support (Postproduction). In addition, our participants taught or learned various practical activities such as yoga, dance sports, aerobic dance and swimming in fifteen universities of Korea, and, we could not match the development phase. We described these in ‘Measure’ (“… In the development phase, the detailed interface design (preproduction), the actual web development (production), and the review and usability testing of the product (postproduction) are included [16]. In our study, the educators selected the primary delivery mode in the design phase and subsequently determined the appropriate tools and the location for production or recording of online class lectures. For real-time lecture, it is very important to provide the most responsive internet experience for the educators and the learners. The educators produced the archives and the bulletin boards for the learner-content interaction and the internet forums for the learner-learner interaction. In case of any difficulties in online classes, the educators or faculties provided the technical support.”) and Table 2.
For these reasons, we did not investigated the analysis phase and development phase using questionnaires because they had already been determined, and we described them in Table 2, and stated that in the ‘Introduction’ of our manuscript. (For example, we described that “In this study, we analysed the OPC conducted during the COVID-19 pandemic in the Republic of Korea according to the ADDIE model of instructional design. First, we suitably modified the ADDIE model for OPC and included questions on design, implementation, and evaluation of the ADDIE model’s components for educators and students.”)
However, according to your comment, we specifically stated them in our ‘Measure’ section of ‘Material and methods’. We represented the specific modifications in response to the comments by blue-letters
For example,
Measures
The analysis phase of the ADDIE model is composed of two levels: needs assessment and front-end analysis. In this phase, educators ascertain the needs of the learners [13], and the purpose of this phase is to determine the performance problems and the objectives of the instruction [16]. Thus, in this study, the need of the learners was the practical exercises, and the front-end analysis was the improvement of physical performance or skills because learners had to take the OPC in PE as their major and/or liberal arts in their Universities. During the design phase, appropriate media and instructional strategies are selected for the objectives identified during the analysis phase [13,16]. Therefore, we selected the web-based delivery (synchronous or asynchronous) as the primary delivery mode of the learning intervention which was the best method under the COVID-19 pandemic. We also analyzed the instruction strategies for the objectives of online PE through questions, such as ‘whether or not to use of various instruction methods and interaction between educator and learner’. The development phase mainly consists of the creation of course materials and instructor guides. In the development phase, the detailed interface design (preproduction), the actual web development (production), and the review and usability testing of the product (postproduction) are included [16]. In our study, the OPC in PE was urgently and inevitably conducted under COVID-19 pandemic, and thus, the educators selected the primary delivery mode in the design phase and subsequently determined the appropriate tools and the location for production or recording of online class lectures. For real-time lecture, it is very important to provide the most responsive internet experience for the educators and the learners. The educators produced the archives and the bulletin boards for the learner-content interaction and the internet forums for the learner-learner interaction. In case of any difficulties in online classes, the educators or faculties provided the technical support. In the implementation phase, educators deliver the instruction, and some support is necessary for successful implementation [13,16]. Finally, formative and summative evaluations are conducted for web-based learning interventions, just as they are for traditional classroom-based learning interventions [16]. In the evaluation phase, educators obtain feedback about their lecture, which can be appropriately addressed in the next online lecture. Thus, in this study, we investigated the characteristics of implementation and evaluation of online-based PE classes. The modified ADDIE model and the questions for online PE classes are presented in Table 2. Finally, we asked the following question in a narrative form: ‘What needs to be added to make the online practical class in PE better? Feel free to describe your opinion.’ Through this open-ended question, we could get loads of information about technical support, overall satisfaction in teaching-learning, and remedy for OPC in PE.
Table 2. Modified ADDIE (Analysis, Design, Development, Implementation, Evaluation) Model and Questions for Online Physical Education Classes.
|
Items |
Components |
Questions to educator |
Questions to learner |
||
|
Analysis* |
Needs analysis |
Practical exercises or activities in physical education including yoga, dance sports, aerobic dance and swimming |
|||
|
Front-end analysis |
Improvement of physical performance or skills |
||||
|
Design |
High-level media selection |
Primary delivery mode of the learning intervention: Web-based delivery (synchronous or asynchronous) |
|||
|
8 |
How was the online physical education class conducted? |
8 |
How was the online physical education class conducted? |
||
|
Objective-level media selection (Instructional strategies for the objectives) |
9 |
Did you use the forums, archives, and bulletin boards online? |
9 |
Have you taken classes online using the forums, archives, and bulletin boards online? |
|
|
10 |
Did you organize and teach team projects or cooperative learning in online practical class? |
10 |
Did a team project or cooperative study take place? |
||
|
11 |
Did you teach using online media such as video editing (Did you appropriately use various techniques when you made video for online lecture) and platform (e.g. Black board to upload video) in online practical class? |
11 |
Was it conducted using online media such as video editing and platform? (Was the lecture video appropriately edited or Were various techniques applied to that lecture video? Could you use the black board to upload your file for interaction with lecturer?) |
||
|
12 |
Did you interact with the learners during the online practical class? |
12 |
Did you interact with the professor during the online practical class? |
||
|
Development* |
Preproduction: |
∙ Pre-recording using appropriate tools and location after selection of the primary delivery mode in the design phase |
|||
|
Production: |
∙ Video editing or Real-time lecture, Internet archives, Bulletin boards, and Internet forums |
||||
|
Postproduction: |
∙ Testing, Review, and Provision of solution for errors or problems |
||||
|
Implementation |
Implementation |
13 |
Was there an advance notice for the online practical class in the context? |
13 |
Was there an advance notice for the online practical class in the context? |
|
14 |
Was the part needed for the online practical class supported? |
14 |
Was the part needed for the online practical class supported? |
||
|
15 |
Did any technical problem, error, or correction occur after the online practical class? |
15 |
Did any technical problem, error, or correction occur after the online practical class? |
||
|
Evaluation |
Evaluation on learning attitude |
16 |
How do you think the learner's physical exercise assignment practice was conducted? |
16 |
How did you practice the physical education project? |
|
Summative |
17 |
Did you conduct the online practical class at the level of difficulty appropriate to the level of each learner? |
17 |
Has the online practical class been conducted at a level suitable for the learner's individual level? |
|
|
18 |
Did you conduct the online practical class as planned? |
18 |
Was the online practical class conducted as planned? |
||
|
19 |
Was the online practical class effectively conducted? |
19 |
Was the online practical class effectively conducted? |
||
|
Formative |
20 |
Did you achieve the class goal you want to achieve through online practical class? |
20 |
Did you achieve the class goal you want to achieve through online practical class? |
|
|
21 |
Compared to face-to-face classes, has the learner's physical performance improved? |
21 |
Compared to the face-to-face class, did you improve your physical skills? |
||
|
22 |
Can you grasp the individual progress of the learners after the online practical class? |
22 |
Were you able to grasp your personal progress after the online practical class? |
||
|
* The analysis and development phases were already determined and were not investigated in this study using questionnaire. They were described as they were actually done. |
|||||
I wish this response can satisfy you, and please explain it further if there was something missing or not fully explained. I appreciate your comment. Owing to your comment, I think that our study will be more improved.
Best Regards,
15 December 2020
